# Non-Effective Improvement of Absorption for Some Nanoparticle Formulations Explained by Permeability under Non-Sink Conditions

**DOI:** 10.3390/pharmaceutics14040816

**Published:** 2022-04-07

**Authors:** Kazuya Sugita, Noriyuki Takata, Etsuo Yonemochi

**Affiliations:** 1Department of Physical Chemistry, Hoshi University, 2-4-41, Ebara, Shinagawa, Tokyo 142-8501, Japan; sugita.kazuya55@chugai-pharm.co.jp; 2Analytical Development Department, Chugai Pharmaceutical Co., Ltd., 5-5-1, Ukima, Kita, Tokyo 115-8543, Japan; takatanry@chugai-pharm.co.jp

**Keywords:** nanoparticle formulation, permeability, non-sink condition, lipophilicity, absorption

## Abstract

We evaluated the in vitro permeability of nanoparticle formulations of high and low lipophilic compounds under non-sink conditions, wherein compounds are not completely dissolved. The permeability of the highly lipophilic compound, griseofulvin, was improved by about 30% due to nanonization under non-sink conditions. Moreover, this permeability was about 50% higher than that under sink conditions. On the other hand, for the low lipophilic compound, hydrocortisone, there was no difference in permeability between micro-and nano-sized compounds under non-sink conditions. The nanonization of highly lipophilic compounds improves the permeability of the unstirred water layer (UWL), which in turn improves overall permeability. On the other hand, because the rate-limiting step in permeation for the low lipophilic compounds is the diffusion of the compounds in the membrane, the improvement of UWL permeability by nanonization does not improve the overall permeability. Based on this mechanism, nanoparticle formulations are not effective for low lipophilic compounds. To accurately predict the absorption of nanoparticle formulations, it is necessary to consider their permeability under non-sink conditions which reflect in vivo conditions.

## 1. Introduction

In recent years, due to the broadening of pharmaceutical targets and the desire for better medicinal efficacy, the chemical structures of drug candidates have become much more complicated [1]. As a result, a lot of drug candidates have the problem of low absorption, caused by poor solubility [1,2,3]. To overcome this issue, various formulation technologies have been developed to improve absorption, such as nanoparticle formulations, amorphous solid dispersions (ASDs) and the use of solubilizing additives [4,5,6]. Among these, nanoparticle formulations have successfully improved the absorption of various pharmaceutical compounds, and several, such as Rapamune^®^, Emed^®^, Tricor^®^, MEGACE^®^ ES, have already been launched as pharmaceuticals.

In pharmaceutical development, the practice of predicting in vivo absorption with in vitro tools is essential for optimizing formulations and designing manufacturing processes [3,7,8,9,10,11,12,13]. However, it is difficult to predict the absorption of nanoparticle formulations using the existing in vitro tools [14]. Though nanoparticle formulations successfully increased the absorption of some compounds by more than ten-fold, they have been unable to increase that of other compounds [15]. The mechanism by which nanoparticle formulations improve absorption can be explained by solubility, dissolution rate and permeability [15,16,17]. The relationship between solubility and particle size is described by the Ostwald–Freundlich equation. Based on this equation, the solubility of small-molecule compounds is calculated to improve by about 10% to 15% when the particle size of the un-milled compounds decreases to 100 nm [16,18]. As many studies have shown that nanoparticle formulations can more than double the absorption of some compounds, we do not think that solubility contributes to the absorption of nanoparticle formulations [15]. The relationship between dissolution rate and particle size is described by the Noyes–Whitney equation [15,16]. For some poorly water-soluble compounds, the rate-limiting step in the absorption is in the dissolution process. It has been reported that nanoparticle formulations improve the absorption of fenofibrate and ritonavir by improving the dissolution rate [19,20]. Recently, it was reported that nanoparticle formulations could also affect permeability [17]. The nanoparticle-improved permeability of poorly water-soluble compounds, namely aprepitant and fenofibrate, has been observed in both in vivo and in vitro models [21,22]. Though these studies suggested that the improved overall permeability was caused by the improved permeability of the unstirred water layer (UWL), the detailed mechanism has not been elucidated.

In this study, we elucidate the mechanism by which nanoparticle formulations are able to improve permeability. The permeation process in passive diffusion is composed of two continuous processes: diffusion of compounds in the UWL, and in the membrane [23,24]. Membrane permeability (P_m(app)_) is not affected by the undissolved compounds. On the other hand, our previous study clarified that UWL permeability (P_UWL(app)_) is affected by them [25]. The undissolved compounds can enter the UWL, resulting in the reduction of the apparent UWL thickness (h_UWL(app)_) and the improvement of P_UWL(app)_. The effectiveness of P_UWL(app)_ improvement will depend on the particle size of the compounds. The nano-sized compounds may improve P_UWL(app)_ much more than the micro-sized compounds. In addition, P_m(app)_ depends on the lipophilicity of the compounds [26]. Therefore, for highly lipophilic compounds, the rate-limiting step in permeation is the diffusion of the compounds in the UWL, and for low lipophilic compounds it is the diffusion of the compounds in the membrane. Thus, nanoparticle formulations may improve the absorption of highly lipophilic compounds, but not low lipophilic compounds. To confirm the improvement of permeability by nanoparticle formulations experimentally, we had to measure permeability under non-sink conditions, wherein compounds are not dissolved completely. However, the conventional Caco-2 and parallel artificial membrane permeability assay (PAMPA) only measures drug permeability under sink conditions, where they are fully dissolved. Thus, the P_UWL(app)_ of nanoparticle formulations has never been reported.

In this study, we prepared nanosuspensions of a highly lipophilic compound, griseofulvin (MW = 352.77 and Log P = 2.18), and a low lipophilic compound, hydrocortisone (MW = 362.46 and Log P = 1.55), by wet milling (Figure 1) [27,28]. We measured the permeability of the un-milled samples and the nanosuspensions under non-sink conditions using the in vitro tool Pion MicroFlux™ (Billerica, MA, USA), which has a permeation compartment similar to that of PAMPA. The purpose of this study was to elucidate the mechanism by which nanoparticle formulations improve permeability by reducing the h_UWL(app)_ and improving the P_UWL(app)_. In addition, we aimed to reveal that the improvement of permeability depends on the lipophilicity of the compounds.

## 2. Materials and Methods

### 2.1. Materials

The reagents were used as received. Griseofulvin was purchased from Tokyo Chemical Industry Co., Ltd. (Tokyo, Japan). The purity of the griseofulvin was guaranteed to be ≥97.0%. Hydrocortisone, methanol, polyvinylpyrrolidone K30 (PVP K30), buffer components (sodium dihydrogen phosphate dihydrate (NaH_2_PO_4_⸱2H_2_O), sodium hydroxide (NaOH), sodium chloride (NaCl)), heptane, sorbitan monooleate, acetonitrile (MeCN) and trifluoroacetic acid (TFA) were purchased from FUJI FILM Wako Pure Chemical Co. (Osaka, Japan). Hydrocortisone and PVP K30 were Wako special grade. The purity of the hydrocortisone was guaranteed to be ≥97.0%. The NaH_2_PO_4_⸱2H_2_O, NaOH, NaCl and heptane were guaranteed reagents. The purity of the NaH_2_PO_4_⸱2H_2_O, NaOH, NaCl and heptane was guaranteed to be 99.0–102.0%, ≥97.0%, ≥99.5% and ≥99.0%. The methanol, MeCN and TFA were for high-performance liquid chromatography (HPLC). The purity of the methanol, MeCN and TFA was guaranteed to be ≥99.7%, ≥99.9% and ≥99.8%. Dioctyl sulfosuccinate (AOT) and sodium lauryl sulfate (SLS) were purchased from Sigma–Aldrich (St. Louis, MO, USA). The purity of the AOT and SLS was guaranteed to be ≥97% and ≥98.0%. Gastrointestinal tract (GIT) lipids and the acceptor sink buffer (ASB) were purchased from Pion Inc. (Billerica, MA, USA).

### 2.2. Methods

#### 2.2.1. Preparation of Microparticles and a Nanosuspension for Griseofulvin

Two sizes of griseofulvin microparticle were prepared. The purchased griseofulvin was the smaller microparticle (S-microparticle griseofulvin). The larger microparticle (L-microparticle griseofulvin) was prepared by recrystallization. The griseofulvin was dissolved in methanol at 70 °C. The solution was cooled to room temperature. The obtained microparticles were isolated by filtration and allowed to dry under the vacuum.

A nanosuspension of griseofulvin (nanosuspension griseofulvin) was prepared by wet milling as described in the previous report [29]. 570 mg of griseofulvin was milled in 5.1 mL of 1.33% PVP K30/0.066% AOT using 24 g of Nikkato YTT 0.8 mm zirconia beads. The milling was performed by a magnetic stirrer at 700 rpm for 30 min and repeated four times. The interval time was 15 min. The zirconia beads were removed using a TERUMO needle syringe (27 gauge) with an inner diameter smaller than the zirconia beads.

#### 2.2.2. Preparation of a Microparticle and a Nanosuspension for Hydrocortisone

The purchased hydrocortisone was used as the microparticle sample (microparticle hydrocortisone). A nanosuspension of hydrocortisone (nanosuspension hydrocortisone) was prepared by wet milling according to a previous report [30,31]. 0.2% PVP K30/0.05% SLS was used as the milling solvent, and the other conditions were the same as those applied to the griseofulvin.

#### 2.2.3. X-ray Powder Diffraction (XRPD)

X-ray powder diffraction (XRPD) patterns in the range of 3° to 35° (2θ) were obtained using a Malvern Panalytical Empyrean powder X-ray diffractometer with Cu Kα radiation in transmission mode. A tube voltage of 45 kV and amperage of 40 mA were used. The nanosuspensions separated by centrifuging and the microparticle samples were used to measure the XRPD patterns.

#### 2.2.4. Particle Size Measurement

The particle size distributions of the nanosuspensions were measured by Malvern Instruments Zetasizer Pro dynamic light scattering (DLS). Before the measurements, the nanosuspensions were diluted to the appropriate concentration using water to prevent multiple scattering. We visually confirmed that the nanosuspensions were dispersed in the water during the measurements. All the measurements were performed in triplicate.

The particle size distributions of the microparticle samples were measured by a Malvern Instruments Mastersizer 3000 laser diffraction size analyzer. Before the measurements, the microparticle samples were suspended in the saturated heptane solution of the compound containing 0.2% sorbitan monooleate which was used to keep the microparticle samples dispersed during the measurements. And, to prevent bubbles in the test samples caused by sorbitan monooleate, heptane was chosen as the solvent instead of water. All the measurements were performed in triplicate.

The morphologies of the nanosuspensions and microparticle samples were visualized using JEOL JSM-IT500HR scanning electron microscopy (SEM). Before the measurement, the mounted samples were coated with platinum under vacuum. We measured different points in more than three images to confirm that the pictured image reflected the whole sample and was not biased in terms of particle size and shape.

#### 2.2.5. Permeability Measurement by MicroFlux™

The microparticle samples were suspended in 1.33% PVP K30/0.066% AOT for griseofulvin and 0.2% PVP K30/0.05% SLS for hydrocortisone. The test sample suspensions in the donor chamber were prepared by diluting the suspensions to a pH 6.5 phosphate buffer. The sample dose amounts of the test sample suspensions were set at 200 μg/mL for griseofulvin and 2000 μg/mL for hydrocortisone to achieve the non-sink conditions. These test sample suspensions were stirred well by the rotation stirrers at 37 °C before the permeability measurements.

The permeability was measured by a Pion MicroFlux™. The measurement conditions were the same as those in our previous study [25]. A polyvinylidene fluoride (PVDF) membrane filter of 0.45 μm pore size with 25 μL of GIT lipids solution was used as a membrane compartment. After 20 mL of ASB was added in each acceptor chamber, 20 mL of the test sample suspension (4 mg of griseofulvin or 40 mg of hydrocortisone) was added in the donor chamber. During the measurement, the solutions in the donor and acceptor chambers were stirred by cross-bar magnetic stirrers at 150 rpm, and were maintained at 37 °C. All permeability measurements were performed in triplicate.

At 0, 30, 60, 120, 240 and 360 min, 100 μL of the acceptor chamber solution was withdrawn and diluted 2-fold. At 0, 120 and 360 min, 1000 μL of the donor chamber solution was withdrawn and filtered through a Cytiva Whatman^®^ Anotop^®^ 10 glass microfiber membrane filter of 0.02 μm pose size (Little Chalfont, UK). To measure the exact concentration in the donor chamber containing the nanosuspensions, the filtration method was applied according to a previous report [32]. Then, 100 μL of the filtered solution was diluted 2-fold. The sample concentration was determined by ultra-high-performance liquid chromatography (UHPLC).

From the obtained concentration–time profiles in the acceptor chambers, the flux (J), which means the mass transfer through the membrane, was calculated by Equation (1):(1)$$J\;\left(t\right)=\frac{1}{A}\cdot \frac{\mathrm{dm}}{\mathrm{dt}}=\frac{V}{A}\cdot \frac{\mathrm{dC}\left(t\right)}{\mathrm{dt}}$$
where dm/dt is the total amount of material crossing the membrane per unit time, A is the area of the membrane (1.54 cm^2^), V is the volume of the acceptor chamber (20 mL), and dC(t)/dt is the slope of the concentration–time profiles in the acceptor chambers. To compensate for the concentration decrease in the donor chamber, and the lag times in the permeation, the initial fluxes from 30 min to 120 min were used to calculate dC(t)/dt. As ASB can keep the acceptor chambers under sink condition during measurement, the flux is described by Equation (2):(2)$$J\;\left(t\right)=P_{\mathrm{app}}C_{D}\left(t\right)$$
where P_app_ is the apparent permeability of the compounds and C_D_ (t) is the concentration of the dissolved compounds in the donor chambers. C_D_ (t) at 0 min (=C_D_ (0)) was used to calculate P_app_ by Equation (2).

#### 2.2.6. UHPLC Analysis

The concentrations of griseofulvin and hydrocortisone were determined using a Waters Acquity UPLC H-Class system and an Acquity UPLC^®^ BEH Shield RP18 1.7 μm, 2.1 × 50 mm column. 0.05%TFA/water (*v*/*v*) and 0.05%TFA/MeCN (*v*/*v*) were used as gradient mobile phase A and gradient mobile phase B respectively. At a flow rate of 1.0 mL/min, the gradient program was initially set as 5% B, and increased to 100% B over 2 min. The column temperature was controlled at 35 °C. The injection volume and ultraviolet (UV) wavelength for griseofulvin were 5 μL and 240 nm, respectively. The injection volume and UV wavelength for hydrocortisone were 1 μL and 254 nm, respectively.

## 3. Results

### 3.1. Characterization of Microparticles and Nanosuspensions

The particle size distributions of the nanosuspensions and microparticle samples for griseofulvin and hydrocortisone are shown in Figure 2. The particle size distribution parameters are summarized in Table 1. Three sizes of test samples were prepared for griseofulvin. The median diameter of the nanosuspension griseofulvin was 0.30 μm, the median diameter of the S-microparticle griseofulvin was 13 μm, and the median diameter of the L-microparticle griseofulvin was 34 μm. Two sizes of hydrocortisone test samples were prepared. The median diameter of the nanosuspension hydrocortisone was 0.25 μm, and the median diameter of the microparticle hydrocortisone was 6.1 μm. In addition, the SEM images of the particles for each sample were consistent with the results of the particle size measurement (Figure 3).

The XRPD patterns of the griseofulvin and hydrocortisone samples are shown in Figure 4. For both griseofulvin and hydrocortisone, the nanosuspensions and the microparticle samples showed the same diffraction pattern, and the positions of the diffraction peaks were not changed by wet milling. The results confirmed that they had the same crystalline form and that wet milling did not cause any polymorphic transition.

### 3.2. Permeability Measurement

#### 3.2.1. Griseofulvin

The griseofulvin time-concentration profiles in the acceptor chambers and the donor chambers, respectively, are shown in Figure 5. The P_app_ of each sample was calculated by Equations (1) and (2) as shown in Figure 6. The samples in the donor chambers were visually confirmed to be suspensions, meaning that they were under non-sink conditions. Based on the concentrations in the donor chambers, we confirmed that the solubility of the griseofulvin was constant, independent of the particle size, and that solubility was not improved by nanonization. In addition, the compounds in the donor chambers were kept in saturation during the measurements. We also confirmed by DLS that the particle size distributions of the nanosuspensions did not change during the measurements. The P_app_ of the nanosuspension griseofulvin was about 30% higher than that of the S-microparticle griseofulvin. On the other hand, the P_app_ of the S-microparticle griseofulvin and the L-microparticle griseofulvin were the same.

#### 3.2.2. Hydrocortisone

The hydrocortisone time-concentration profiles in the acceptor chambers and donor chambers are shown in Figure 7. The P_app_ of each sample was calculated by Equations (1) and (2) as shown in Figure 8. The non-sink condition of the donor chamber samples, the saturation of the compounds in the donor chambers, and the consistent particle size distributions of the nanosuspensions during the measurements were confirmed for both hydrocortisone samples as well as for the griseofulvin. The solubility of the hydrocortisone was not improved by nanonization. The P_app_ of the nanosuspension hydrocortisone was almost the same as that of the microparticle hydrocortisone. Therefore, the nanoparticle formulation did not improve the permeability of the hydrocortisone.

#### 3.2.3. Calculation of P_m(app)_ and P_UWL(app)_

As the griseofulvin and the hydrocortisone were neutral compounds, the molecules in the donor compartment were unionized free compounds or undissolved compounds. In this condition, the intrinsic membrane permeability was equal to the P_m(app)_.

The P_UWL(app)_ of the griseofulvin was theoretically calculated by the following equation [25]:(3)$$\frac{1}{P_{\mathrm{app}}}=\frac{1}{P_{m\;\left(\mathrm{app}\right)}}+\frac{1}{P_{\mathrm{UWL}\;\left(\mathrm{app}\right)}}$$

In our previous study, the P_m(app)_ of the griseofulvin was 0.0284 cm/min [25]. P_m(app)_ does not depend on the sink/non-sink conditions or the particle size of compounds. By substituting the measured P_app_ in Equation (3), the P_UWL(app)_ of the griseofulvin for each test sample was calculated as shown in Table 2. The P_UWL(app)_ of the nanosuspension griseofulvin was almost twice as fast as that of the S-microparticle griseofulvin.

As the P_m(app)_ of hydrocortisone has never been reported, we theoretically calculated the impact of the P_UWL(app)_ on the P_app_ of the hydrocortisone using the h_UWL(app)_ under non-sink conditions. Both the P_app_ of the nanosuspension hydrocortisone and the microparticle hydrocortisone were 0.00239 cm/min. According to Fick’s first law, P_UWL(app)_ is calculated using the apparent aqueous diffusivity (D_aq(app)_) and the h_UWL(app)_ as follows:(4)$$P_{\mathrm{UWL}\;\left(\mathrm{app}\right)}=\frac{D_{\mathrm{aq}\;\left(\mathrm{app}\right)}}{h_{\mathrm{UWL}\;\left(\mathrm{app}\right)}}$$

D_aq(app)_ can be empirically estimated using the following equation [33] (p. 381):(5)$${\mathrm{Log}\;D}_{\mathrm{aq}\;\left(\mathrm{app}\right)}=-4.131-0.4531\mathrm{Log}\;\mathrm{MW}$$

The D_aq(app)_ of the hydrocortisone was calculated to be 5.12 × 10^−6^ cm^2^/s by Equation (5). The h_UWL(app)_ under sink conditions was reported to be about 100 μm [23]. Therefore, we assumed that the h_UWL(app)_ under non-sink conditions was less than 100 μm. When the h_UWL(app)_ was from 1 μm to 100 μm, the P_UWL(app)_ of the hydrocortisone was calculated as shown in Table 3. The calculated P_m(app)_ was much smaller than the calculated P_UWL(app)_ in both conditions. Therefore, the diffusion of the compounds in the membrane was the rate-limiting step in the permeation of the hydrocortisone.

## 4. Discussion

These results demonstrate the mechanism we proposed in the introduction to explain how nanoparticle formulations improve permeability. The highly lipophilic compound, griseofulvin, had high permeability, and nanonization increased its P_app_ by about 30%. On the other hand, the low lipophilic compound, hydrocortisone, had low permeability and nanonization did not change its P_app_. In this section, we discuss the importance of permeability under non-sink conditions in predicting the absorption of nanoparticle formulations. We also discuss whether nanoparticle formulations can improve the permeability of small molecule compounds which have various lipophilicities.

To assess the impact of permeability under non-sink conditions, we compared it with that under sink conditions. The P_app_ of griseofulvin under sink conditions was 0.0148 cm/min in our previous study [25]. Therefore, the nanoparticle formulation of griseofulvin improved permeability by about 50% under non-sink conditions compared to that under sink conditions. (Figure 9) This effect under non-sink conditions has a significant impact on the absorption of nanoparticle formulations.

Using the results of permeability measurements, we quantitatively estimated the impact of nanoparticle formulations on permeability for small molecule compounds. As described in the Results section, the h_UWL(app)_ was reported to be around 100 μm under sink conditions [23]. Under non-sink conditions, the h_UWL(app)_ was calculated by the following equation, using D_aq(app)_, P_app_ and P_m(app)_ [25]:(6)$$h_{\mathrm{UWL}\;\left(\mathrm{app}\right)}=D_{\mathrm{aq}\;\left(\mathrm{app}\right)}\left(\frac{1}{P_{\mathrm{app}}}-\frac{1}{P_{m\;\left(\mathrm{app}\right)}}\right)$$

The D_aq(app)_ of griseofulvin was 5.18 × 10^−6^ cm^2^/s and the P_m(app)_ was 0.0284 cm/s. Substituting the measured P_app_ into Equation (6), the h_UWL(app)_ was calculated to be around 70 μm for the microparticle samples and around 30 μm for the nanosuspensions under non-sink conditions (Figure 10). The nanonization of particle size significantly reduced the h_UWL(app)_. When compounds are nanonized, the undissolved compounds can deeply penetrate the UWL, causing it to shrink. Considering this h_UWL(app)_ reduction mechanism, the micronization of griseofulvin may have improved the P_app_. However, the S-microparticle griseofulvin and the L-microparticle griseofulvin did not show a clear difference in P_app_. Similar results showing the effect of micronization on P_app_ were reported previously in vivo, but the mechanism has so far not been investigated in detail [34]. To understand how micronization improves permeability, further research is necessary.

To evaluate the relationship between the lipophilicity of compounds and the improvement of P_app_ by nanonization, the P_app_ of microparticle and nanoparticle compounds with P_m(app)_ ranging from 0.001 cm/min to 1.0 cm/min was calculated under sink and non-sink conditions (Figure 11). The P_app_ can be calculated using P_m(app)_, h_UWL (app)_ and D_aq(app)_ as described in Equation (7) [25]:(7)$$P_{\mathrm{app}}=\frac{1}{P_{m\;\left(\mathrm{app}\right)}}-\frac{h_{\mathrm{UWL}\;\left(\mathrm{app}\right)}}{D_{\mathrm{aq}\;\left(\mathrm{app}\right)}}$$

The D_aq(app)_ of the model compounds (MW = ca.350) was calculated to be 5.20 × 10^−6^ cm^2^/s using Equation (5). Based on the measurements of griseofulvin permeability, the h_UWL(app)_ was estimated to be 100 μm under sink conditions, 70 μm for microparticle compounds under non-sink conditions, and 30 μm for nanoparticle compounds under non-sink conditions. The lipophilicity and membrane permeability of griseofulvin were Log P = 2.18 and P_m(app)_ = 0.0284 cm/min, respectively. When the P_m(app)_ of the compound is 0.0284 cm/min, its P_app_ under non-sink conditions for nanoparticle compounds was estimated to be about 50% higher than that under sink conditions. On the other hand, the lipophilicity of hydrocortisone is Log P = 1.55 and its membrane permeability was estimated to be P_m(app)_ = 0.00259 cm/min. When the P_m(app)_ of the compound was 0.00259 cm/min, its P_app_ under non-sink conditions for nanoparticle compounds was estimated to be almost the same as that under sink conditions. These results were consistent with the results of the permeability measurements. In addition, the effect of nanonization on the P_app_ of the higher lipophilic compound was also calculated. The lipophilicity of the neutral compound, progesterone, was Log P = 3.48 and its P_app_ using PAMPA was reported to be 2–3 times higher than that of griseofulvin [35]. When the P_app_ of the compound under sink conditions was twice as high as that of griseofulvin (P_app_ = 0.030 cm/min), its P_m(app)_ was about 1.0 cm/min. Based on Figure 11, nanonization will increase its P_app_ by about three times. Nanonization has previously been reported to increase the in vivo permeability of several highly lipophilic compounds by around 2–3 times [17]. Therefore, the estimation in Figure 11 is reasonable.

The in vivo absorption of compounds is proportional to their concentration and permeability in the human small intestine. Based on Figure 11, nanoparticle formulations greatly improve the permeability of highly lipophilic compounds. This improved permeability can significantly improve absorption. On the other hand, nanoparticle formulations have a very small effect on the permeability of low lipophilic compounds. As the effect of nanonization on solubility is also negligible, nanoparticle formulations will not improve the absorption of low lipophilic compounds. Our results explain the big differences in absorption for each compound. Thus, knowledge of the relationship between lipophilicity and permeability improvement (Figure 11) is critical for developing formulations for poorly water-soluble compounds.

Our proposed mechanism for how nanoparticle formulations improve permeability is illustrated in Figure 12. It explains why nanoparticle formulations improve the permeability of highly lipophilic compounds, but not low lipophilic compounds. Compared with sink conditions, h_UWL(app)_ under non-sink conditions is smaller, becoming even smaller as particle size decreases. The reduction of h_UWL(app)_ by nanonization greatly improves P_UWL(app)_. For highly lipophilic compounds, P_m(app)_ is equal to or larger than P_UWL(app)_, and the improvement of P_UWL(app)_ will in turn improve the P_app_ (Figure 12A). On the other hand, for the low lipophilic compounds, P_m(app)_ is smaller than P_UWL(app)_, and the improvement of P_UWL(app)_ will not improve the P_app_ (Figure 12B). The results of the permeability measurements successfully demonstrated the mechanism described in Figure 12.

We can quantitatively evaluate the impact of nanonization on permeability by measuring permeability under non-sink conditions, enabling the more accurate prediction of absorption. Permeability measured by PAMPA, which was used in the permeation compartment for MicroFlux™, is highly correlated with human jejunal permeability, but only for compounds that are not affected by transporters [35,36]. Though MicroFlux™ and the PAMPA system are apparently different, the mechanism underlying the measurement of the passive diffusion is the same, and important features, such as membrane composition and the donor/acceptor chambers, are similar, suggesting that the permeability measured using MicroFlux™ can also be correlated with human jejunal permeability. Several studies have also reported that MicroFlux™ results are well-correlated with in vivo absorption [22,23,37,38], and that P_UWL(app)_ affects permeability in the human small intestine [39,40]. Therefore, we concluded that the improvement of permeability by nanonization observed in this study can be correlated with permeability in in vivo contexts.

## 5. Conclusions

This is the first report to quantitatively study the impact of nanoparticle formulations on permeability. Furthermore, it was not clear what type of compounds could have their absorption improved by nanoparticle formulations. Thus, we proposed a mechanism explaining how nanoparticle formulations improve permeability. We successfully demonstrated this mechanism by measuring permeability under non-sink conditions, wherein compounds are not completely dissolved. We found that nanoparticle formulations of highly lipophilic compounds can greatly improve permeability under non-sink conditions, resulting in the improvement of absorption. Nanoparticle formulations will improve the absorption of compounds with Log P ≥ ca.3.5 more than three-fold. On the other hand, they do not effectively improve the permeability of low lipophilic compounds. Nanoparticle formulations will not improve the absorption of compounds with Log P ≤ ca.1.5. Based on these findings, the absorption of nanoparticle formulations is affected by both permeability and dissolution. Furthermore, our research suggests that we should investigate technologies other than nanoparticle formulations for low lipophilic compounds. It is critical that permeability is measured under conditions that reflect clinical characteristics, such as dosage and non-sink/sink conditions, in order to accurately predict the absorption of nanoparticle formulations.

## Figures and Tables

**Figure 1 pharmaceutics-14-00816-f001:**
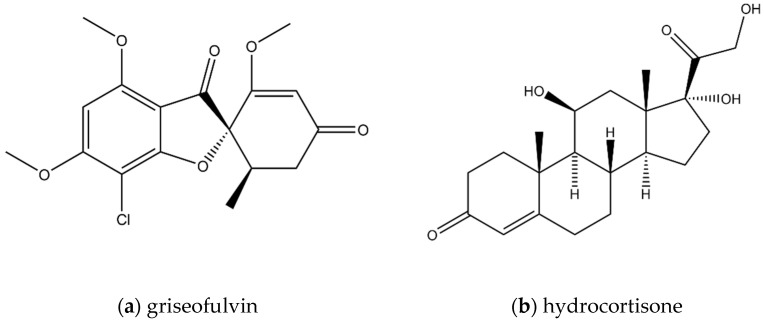
The molecular structure of: (**a**) griseofulvin; (**b**) hydrocortisone.

**Figure 2 pharmaceutics-14-00816-f002:**
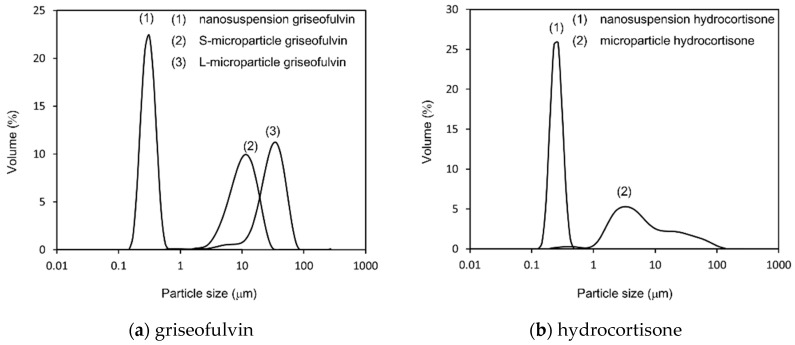
Particle size distribution (PSD) of: (**a**) griseofulvin samples; (**b**) hydrocortisone samples. The results were transformed to volume distribution. The PSD parameters for each sample are summarized in Table 1. The PSD of the nanosuspensions was measured by dynamic light scattering, where water was used as the solvent. The PSD of the microparticle samples was measured by laser diffraction size analyzer, where the saturated heptane solution of the compound containing 0.2% sorbitan monooleate was used as the solvent.

**Figure 3 pharmaceutics-14-00816-f003:**
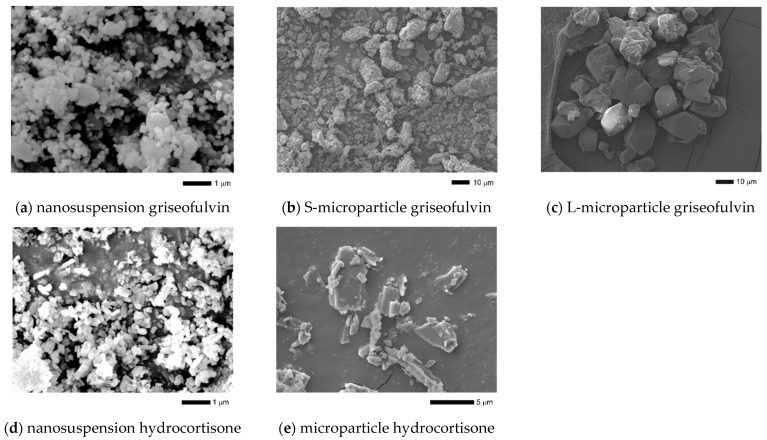
Scanning electron microscopy (SEM) image of griseofulvin samples (**a**–**c**) and hydrocortisone samples (**d**,**e**).

**Figure 4 pharmaceutics-14-00816-f004:**
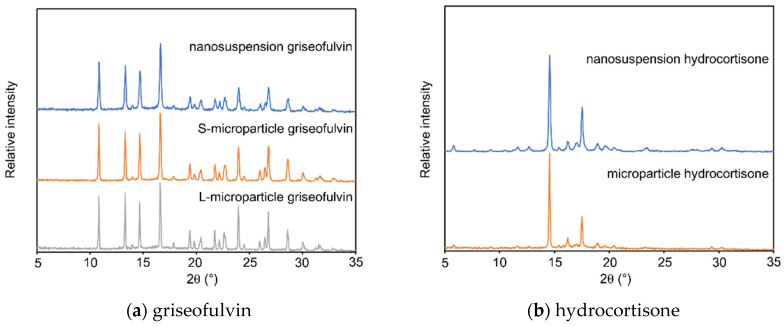
X-ray powder diffraction pattern of: (**a**) griseofulvin samples; (**b**) hydrocortisone samples.

**Figure 5 pharmaceutics-14-00816-f005:**
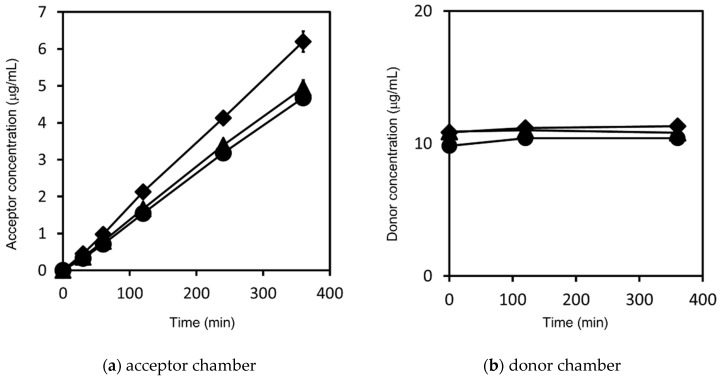
Griseofulvin concentration–time profiles in: (**a**) the acceptor chamber; (**b**) the donor chamber. The results represent the average griseofulvin concentration ± SD (*n* = 3) for nanosuspension griseofulvin (◆), S-microparticle griseofulvin (▲) and L-microparticle griseofulvin (●).

**Figure 6 pharmaceutics-14-00816-f006:**
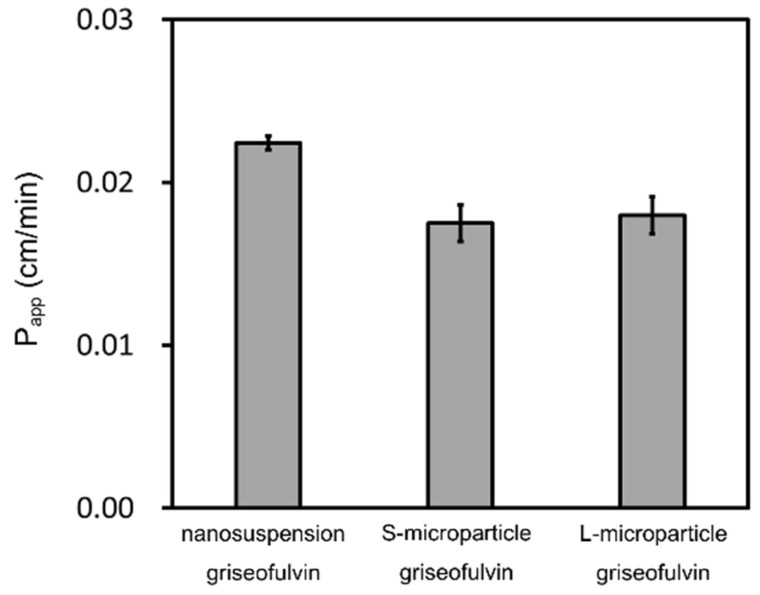
Calculated apparent permeability (P_app_) of griseofulvin. The results represent the average P_app_ ± SD (*n* = 3).

**Figure 7 pharmaceutics-14-00816-f007:**
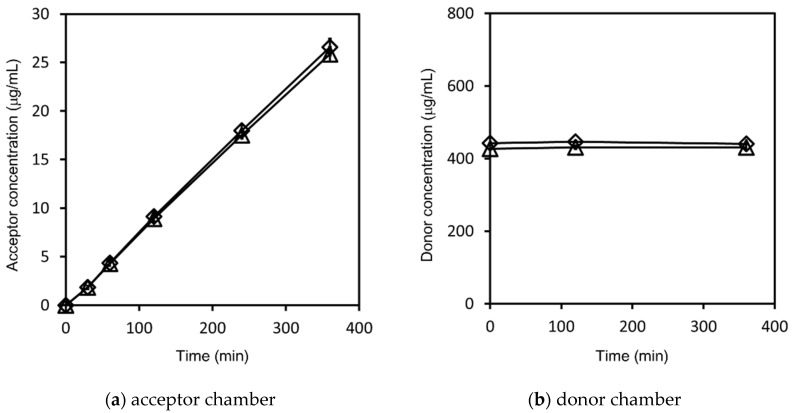
Hydrocortisone concentration–time profile in: (**a**) the acceptor chamber; (**b**) the donor chamber. The results represent the average hydrocortisone concentration ± SD (*n* = 3) for nanosuspension hydrocortisone (◇) and microparticle hydrocortisone (△).

**Figure 8 pharmaceutics-14-00816-f008:**
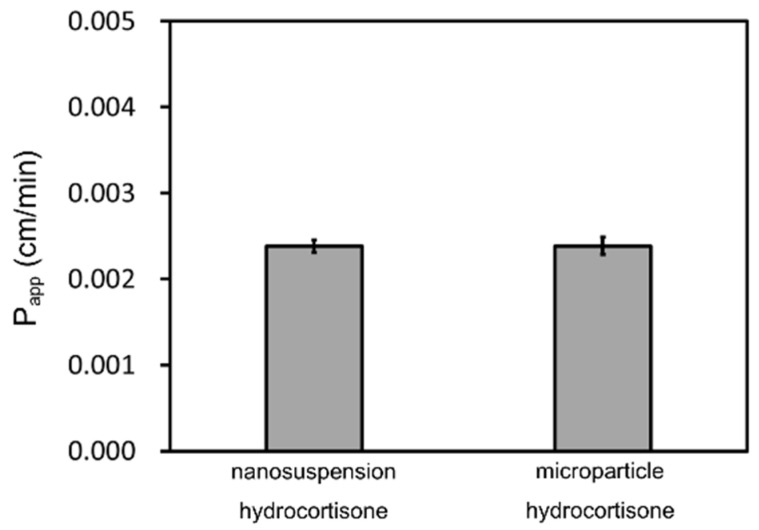
Calculated P_app_ of hydrocortisone. The results represent the average P_app_ ± SD (*n* = 3).

**Figure 9 pharmaceutics-14-00816-f009:**
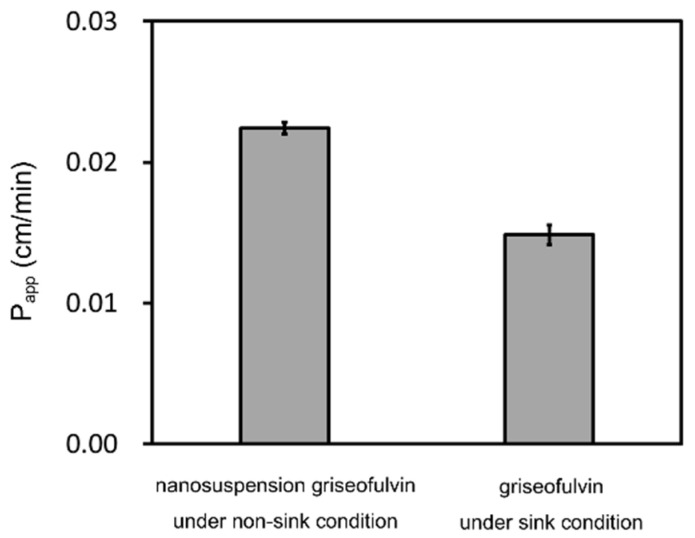
Comparison of P_app_ of nanosuspension griseofulvin under non-sink conditions and sink conditions. The sink condition result is taken from our previous report [25]. The results represent the average P_app_ ± SD (*n* = 3). Adapted from [25], MDPI, 2021.

**Figure 10 pharmaceutics-14-00816-f010:**
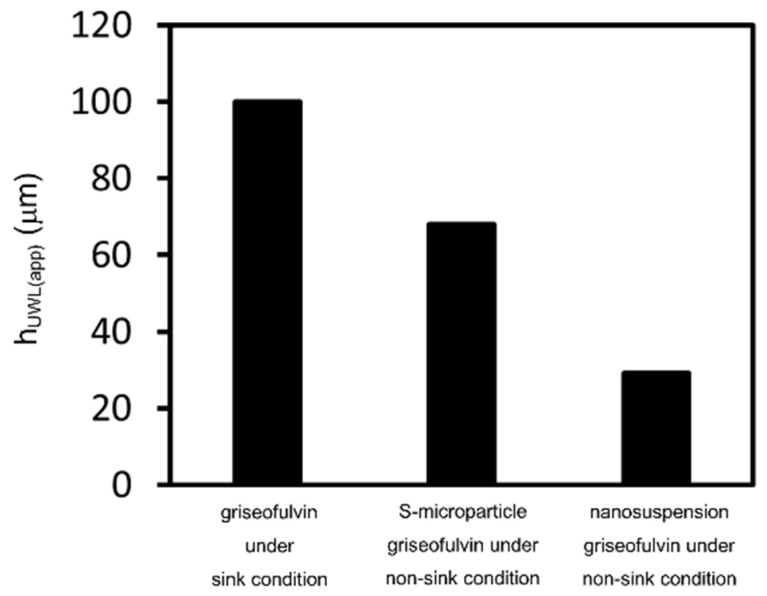
Calculated unstirred water layer (UWL) thickness under sink and non-sink conditions for microparticle samples and for nanosuspensions.

**Figure 11 pharmaceutics-14-00816-f011:**
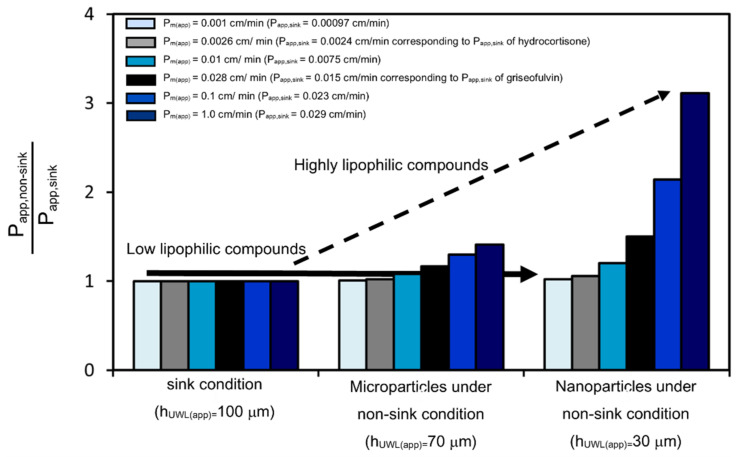
Calculated permeability improvement effect for microparticle and nanoparticle compounds under non-sink conditions. The permeability of model compounds ranges from P_app,sink_ = 0.00097 cm/min to P_app,sink_ = 0.029 cm/min. P_app,sink_ and P_app,non-sink_ represent the P_app_ under sink conditions and under non-sink conditions, respectively.

**Figure 12 pharmaceutics-14-00816-f012:**
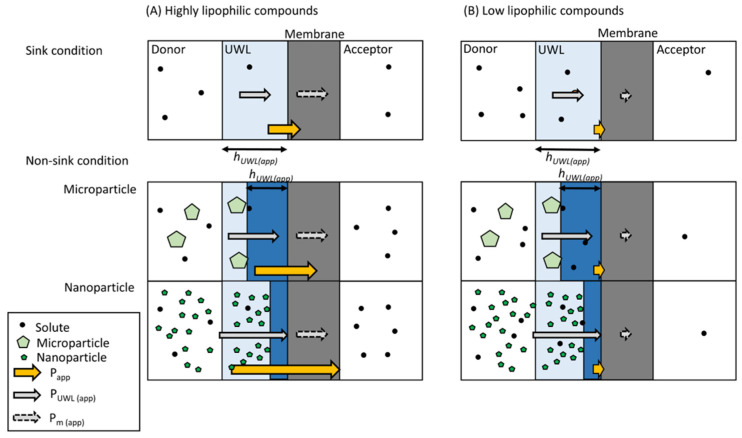
Proposed mechanism of permeability improvement by nanoparticle formulations.

**Table 1 pharmaceutics-14-00816-t001:** PSD parameters of griseofulvin and hydrocortisone samples.

Samples	Methods/Solvents	D10 (μm) ^1,2^	D50 (μm) ^1,2^	D90 (μm) ^1,2^
nanosuspension griseofulvin	dynamic light scattering (DLS)/water	0.22 ± 0.00	0.30 ± 0.00	0.43 ± 0.01
S-microparticle griseofulvin	laser diffraction size analyzer/heptane containing 0.2% sorbitan monooleate ^3^	4.2 ± 1.1	13 ± 1	30 ± 9
L-microparticle griseofulvin	laser diffraction size analyzer/heptane containing 0.2% sorbitan monooleate ^3^	16 ± 1	34 ± 2	60 ± 5
nanosuspension hydrocortisone	DLS/water	0.19 ± 0.00	0.25 ± 0.01	0.33 ± 0.00
microparticle hydrocortisone	laser diffraction size analyzer/heptane containing 0.2% sorbitan monooleate ^3^	2.4 ± 0.5	6.1 ± 0.7	35 ± 3

^1^ Results represent average value ± standard deviation (SD) (*n =* 3). ^2^ D10, D50 and D90 describe the diameter of particles at which 10%, 50% and 90% of the sample particles, respectively, are smaller than those values based on a volume distribution. ^3^ The saturated heptane solution of the compound containing 0.2% sorbitan monooleate was used as the solvent.

**Table 2 pharmaceutics-14-00816-t002:** Calculated permeability of griseofulvin.

Parameter	Nanosuspension Griseofulvin	S-Microparticle Griseofulvin	L-Microparticle Griseofulvin
Measured P_app_ (cm/min)	0.0224	0.0175	0.0180
P_m(app)_ (cm/min)	0.0284		
Calculated P_UWL(app)_ (cm/min)	0.107	0.0456	0.0490

**Table 3 pharmaceutics-14-00816-t003:** Calculated permeability of hydrocortisone.

Parameter				
Measured P_app_ (cm/min)	0.00239			
h_UWL(app)_ (μm)	100	50	10	1
Calculated P_UWL(app)_ (cm/min)	0.0307	0.0615	0.307	3.07
Calculated P_m(app)_ (cm/min)	0.00259	0.00249	0.00241	0.00239

## Data Availability

The data presented in this study are available in this article.

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
