# Peer review of "Non-Effective Improvement of Absorption for Some Nanoparticle Formulations Explained by Permeability under Non-Sink Conditions"

_pharmaceutics, 2022, doi:10.3390/pharmaceutics14040816_

Round 1

Reviewer 1 Report

Introduction. Please remove all equations in the section 2. Materials and Methods.

What about the Figure 1? Is original? If yes, please move this information in the Discussion part. Please convert the Table 1 in text.

Please correct “µ” symbol during the entire manuscript.

Section 2.2.5 and 2.2.6 should be more compact.

Section 3.1. XRD analysis. Please elaborate.

Conclusions need to be improved by specifying the discussed important points within this work. The conclusion section should have the main results in quantitative statements as well.

Summarizing, the manuscript may be interesting but it needs some improvements before it will be accepted to print in this journal.

Author Response

Thank you for your comments on our paper. They have helped us to improve the paper. We have answered each of your points. Please see the attachment.

Reviewer 2 Report

This manuscript determined the increased permeability of suspensions of different particle sizes and different lipophilicities.  The results showed the increased permeability of the nanosized lipophilic drug through the unstirred water layer.  The research could help better understand how nanosuspensions can improve oral absorption.  Generally, the experiments were properly designed and data are clearly presented.  Specific comments are:

  1. Particle size and size distribution was determined before the permeation study. Were they monitored during and after the study?  Could they change during the study period?
  2. Line 157: what were the amounts of drug loaded to the donor chamber?
  3. Equation 4: what concentration was used to calculate permeability, solubility or concentration of drug loaded in the donor (i.e., dissolved and undissolved)?
  4. Table 2: the units for particle sizes are not in “mm”. There are also many typos in the units, e.g., “mm” in the thickness of the UWL, “g/mL” in concentrations, and  “L” in volume.
  5. Tables 3 and 4 could be combined.
  6. The Pm of griseofulvin was determined previously (line 257) but the Pm of hydrocortisone was calculated (Table 4). Were the calculated and experimental values consistent?
  7. How was Equation 9 derived?
  8. It may be considered to add a theory section to discuss the theories and the equations as well as how they were derived and/or applied in the manuscript.

Author Response

(The authors gave the same response as above.)

Reviewer 3 Report

This paper evaluated the in vitro permeability of nanoparticle formulations of high and low lipophilic compounds under non-sink conditions, which makes sense for drug development.

In Table 2, the unit mm should be mm.

Author Response

(The authors gave the same response as above.)

Reviewer 4 Report

Dear Editor,

I accurately reviewed the article

Non-effective improvement of absorption for some nanoparticle formulations explained by permeability under non-sink  conditions

submitted to Pharmaceutics.

In this article the authors study the in vitro permeability of nanoparticle formulations of high and low lipophilic compounds under non-sink conditions, wherein compounds are not completely dissolved. The permeability of the highly lipophilic compound, griseofulvin, was improved by about 30% due to nanonization under non-sink conditions. Moreover, this permeability was about 50% higher than that under sink conditions. On the other hand, for the low lipophilic compound, hydrocortisone, there was no difference in permeability between micro-and nano-sized compounds under non-sink conditions. The nanonization of highly lipophilic compounds improves the permeability of the unstirred water layer (UWL), which in turn improves overall permeability. On the other hand, because the rate-limiting step in permeation for the low lipophilic compounds is the diffusion of the compounds in the membrane, the improvement of UWL permeability by nanonization does not improve the overall permeability. Based on this mechanism, nanoparticle formulations are not effective for low lipophilic compounds.

The topic is suitable for the journal, but the authors need to improve some important aspects and solve some problems.

Introduction

it would be useful to readers if the authors did a broad picture of nanoformulations, citing some recent work:

  1. Nanoformulation and encapsulation approaches for poorly water-soluble drug nanoparticles; Nanoscale, 2016,8, 1746-1769 https://doi.org/10.1039/C5NR07161E
  2. Nanoformulations for Drug Delivery: Safety, Toxicity, and Efficacy; Methods Mol Biol . 2018; 1800:347-365. doi: 10.1007/978-1-4939-7899-1_17.
  3. Directly Resveratrol immobilization on hydrophilic charged gold nanoparticles: structural investigations and cytotoxic studies; Nanomaterials 2020, 10(10), 1898; https://doi.org/10.3390/nano10101898
  4. Nanoformulation Development to Improve the Biopharmaceutical Properties of Fisetin Using Design of Experiment Approach; Molecules 2021 19; 26:3031. doi: 10.3390/molecules26103031.
  5. Highly hydrophilic gold nanoparticles as carrier for anticancer copper(I) complexes: loading and release studies for biomedical applications; Nanomaterials 2019, 9, 772; doi:10.3390/nano9050772

Experimental part

Authors must specify the purity and of the chemical reagents used also to ensure the reproducibility of the experiments.

Before being observed under the SEM, how were the particles prepared? have they been metallized? must be specified.

Moreover, for the SEM study it is important to have many images to be sure of the significance of the data. How many SEM images were taken for each sample? It should be specified.

Results and Discussion

It would be helpful for the authors to explain and discuss the choice of solvents for DLS measurements

Again, about the DLS measurements, the "Malvern Instruments Mastersizer 3000 laser diffraction size analyzer" instrument should have a measurement range from 10 nm to 3.5 mm. How do the authors get the values in table 2 that are greater?

Conclusions

In my opinion the conclusions are too general and unclear. The authors should extrapolate in the conclusions the advantages of studies, comparing with others.

English

The English language must be deeply revised: some sentences are too long and there are many typos.

In conclusion, the article could be suitable for publication, but only after major revisions.

best regards

Author Response

(The authors gave the same response as above.)

Round 2

Reviewer 1 Report

The authors addressed most of the comments and questions I formulated during the first round of revision. The revised version of the manuscript is more complete and readable. In this form the manuscript is suitable for publication to me.

Author Response

Thank you again for your careful review. It was helpful for us.

Reviewer 4 Report

Dear Editor,

I accurately reviewed the article

Manuscript ID pharmaceutics-1633719

Non-effective improvement of absorption for some nanoparticle formulations explained by permeability under non-sink conditions

submitted to Pharmaceutics.

In this article the authors study the in vitro permeability of nanoparticle formulations of high and low lipophilic compounds under non-sink conditions, wherein compounds are not completely dissolved. The permeability of the highly lipophilic compound, griseofulvin, was improved by about 30% due to nanonization under non-sink conditions. Moreover, this permeability was about 50% higher than that under sink conditions. On the other hand, for the low lipophilic compound, hydrocortisone, there was no difference in permeability between micro-and nano-sized compounds under non-sink conditions. The nanonization of highly lipophilic compounds improves the permeability of the unstirred water layer (UWL), which in turn improves overall permeability. On the other hand, because the rate-limiting step in permeation for the low lipophilic compounds is the diffusion of the compounds in the membrane, the improvement of UWL permeability by nanonization does not improve the overall permeability. Based on this mechanism, nanoparticle formulations are not effective for low lipophilic compounds.

The authors have improved some aspects, but many problems remain.

Regarding the DLS, the problem is not only the unit of measurement in microns but the values in the table, which are also in table 1, for D90 also reach diameter values of 29 and 59 and 34, as also commented on the nice text

Experimental part

Details of the reagents and solvents used (the degree of purity) are still missing.

The authors have improved some aspects but the problems remain.

The SEM images provided do not show regular morphologies and dimensions: as it can be said that "there was no difference in size and the shape of the samples. "(line 169) ?

Results and Discussion

It would be helpful for the authors to explain and discuss the choice of solvents for DLS measurements

Again, about the DLS measurements (lines 158-166)

the authors write

"The particle size distributions of the nanosuspensions were measured by a Malvern 158

Zetasizer Pro Dynamic Light Diffusion (DLS) tools. Before the measurements, the 159

the nanosuspensions were diluted to the appropriate concentration using water to prevent 160

multiple dispersion. All measurements were performed in triplicate. 161

The particle size distributions of the microparticle samples were measured by a Malvern 162

Instruments Mastersizer 3000 laser diffraction size analyzer. Prior to measurements, the 163

Microparticle samples were suspended in the saturated heptane solution containing 164

0.2% sorbitan monooleate which kept samples dispersed during measurement - 165

minds. All measurements were performed in triplicate. "

so they use water in one case heptane + 0.2% sorbitan in the other: I had a brief discussion on this aspect.

Figure 2 shows some DLS measurements: are they referred to or traceable to samples in table 1? Which? in which solvent are the measurements made?  it is not clear

The caption of table 1 is not self-consistent: what is meant by D10, D50 D90? in which solvent are the measurements made?

In conclusion, the article could be suitable for publication, but after these major revisions.

best regards

Author Response

(The authors gave the same response as above.)

Round 3

Reviewer 4 Report

Dear Editor,

I accurately reviewed the article

Manuscript ID pharmaceutics-1633719-v3

Non-effective improvement of absorption for some nanoparticle formulations explained by permeability under non-sink conditions

submitted to Pharmaceutics.

In this article the authors study the in vitro permeability of nanoparticle formulations of high and low lipophilic compounds under non-sink conditions, wherein compounds are not completely dissolved. The permeability of the highly lipophilic compound, griseofulvin, was improved by about 30% due to nanonization under non-sink conditions. Moreover, this permeability was about 50% higher than that under sink conditions. On the other hand, for the low lipophilic compound, hydrocortisone, there was no difference in permeability between micro-and nano-sized compounds under non-sink conditions. The nanonization of highly lipophilic compounds improves the permeability of the unstirred water layer (UWL), which in turn improves overall permeability. On the other hand, because the rate-limiting step in permeation for the low lipophilic compounds is the diffusion of the compounds in the membrane, the improvement of UWL permeability by nanonization does not improve the overall permeability. Based on this mechanism, nanoparticle formulations are not effective for low lipophilic compounds.

The authors improved the manuscript and resolved doubts and problems.

In my opinion the manuscript is ready for publication

best regards